# MicroRNA-204 Regulates Angiogenesis and Vasculogenic Mimicry in CD44+/CD24− Breast Cancer Stem-like Cells

**DOI:** 10.3390/ncrna10010014

**Published:** 2024-02-09

**Authors:** Martha Resendiz-Hernández, Alejandra P. García-Hernández, Macrina B. Silva-Cázares, Rogelio Coronado-Uribe, Olga N. Hernández-de la Cruz, Lourdes A. Arriaga-Pizano, Jessica L. Prieto-Chávez, Yarely M. Salinas-Vera, Eloisa Ibarra-Sierra, Concepción Ortiz-Martínez, César López-Camarillo

**Affiliations:** 1Posgrado en Ciencias Genómicas, Universidad Autónoma de la Ciudad de México, CDMX 03100, Mexico; martha.resendiz@estudiante.uacm.edu.mx (M.R.-H.); paola_75_19@hotmail.com (A.P.G.-H.); ediacara79@yahoo.com.mx (O.N.H.-d.l.C.); yarely.vera.ms@gmail.com (Y.M.S.-V.); 2Unidad Academica Multidisciplinaria Región Altiplano, Universidad Autónoma de San Luis Potosí, Matehuala 78760, San Luis Potosí, Mexico; macrina.silva@uaslp.mx; 3Unidad de Investigación Médica en Inmunoquímica, Hospital de Especialidades del Centro Médico Siglo XXI, Instituto Mexicano del Seguro Social, CDMX 06720, Mexico; landapi@hotmail.com; 4Laboratorio de Citometría de Flujo, Centro de Instrumentos, Coordinación de Investigación en Salud, Hospital de Especialidades del Centro Médico Siglo XXI, Instituto Mexicano del Seguro Social, CDMX 06720, Mexico; lakshmi.litmus@hotmail.com; 5Departamento de Investigación, Instituto Estatal de Cancerologia “Dr. Arturo Beltrán Ortega”, Acapulco 39610, Guerrero, Mexico; comite.etica@cancerologiagro.gob.mx; 6Servicio de Ginecología Oncológica, Instituto Estatal de Cancerologia “Dr. Arturo Beltrán Ortega”, Acapulco 39610, Guerrero, Mexico; cortima_@hotmail.com

**Keywords:** breast cancer, cancer stem-like cells, microRNA-204, angiogenesis, vasculogenic mimicry

## Abstract

Tumors have high requirements in terms of nutrients and oxygen. Angiogenesis is the classical mechanism for vessel formation. Tumoral vascularization has the function of nourishing the cancer cells to support tumor growth. Vasculogenic mimicry, a novel intratumoral microcirculation system, alludes to the ability of cancer cells to organize in three-dimensional (3D) channel-like architectures. It also supplies the tumors with nutrients and oxygen. Both mechanisms operate in a coordinated way; however, their functions in breast cancer stem-like cells and their regulation by microRNAs remain elusive. In the present study, we investigated the functional role of microRNA-204 (miR-204) on angiogenesis and vasculogenic mimicry in breast cancer stem-like cells. Using flow cytometry assays, we found that 86.1% of MDA-MB-231 and 92% of Hs-578t breast cancer cells showed the CD44+/CD24− immunophenotype representative of cancer stem-like cells (CSCs). The MDA-MB-231 subpopulation of CSCs exhibited the ability to form mammospheres, as expected. Interestingly, we found that the restoration of miR-204 expression in CSCs significantly inhibited the number and size of the mammospheres. Moreover, we found that MDA-MB-231 and Hs-578t CSCs efficiently undergo angiogenesis and hypoxia-induced vasculogenic mimicry in vitro. The transfection of precursor miR-204 in both CSCs was able to impair the angiogenesis in the HUVEC cell model, which was observed as a diminution in the number of polygons and sprouting cells. Remarkably, miR-204 mimics also resulted in the inhibition of vasculogenic mimicry formation in MDA-MB-231 and Hs-578t CSCs, with a significant reduction in the number of channel-like structures and branch points. Mechanistically, the effects of miR-204 were associated with a diminution of pro-angiogenic VEGFA and β-catenin protein levels. In conclusion, our findings indicated that miR-204 abrogates the angiogenesis and vasculogenic mimicry development in breast cancer stem-like cells, suggesting that it could be a potential tool for breast cancer intervention based on microRNA replacement therapies.

## 1. Introduction

In diverse human malignancies, there exists a specific and under-represented subpopulation of tumor cells that exhibit biological and molecular features similar to those of normal stem cells, known as cancer stem-like cells (CSCs) [1]. They were first recognized in acute myeloid leukemia in 1997 as a subpopulation of tumor-initiating cells originating from primitive hematopoietic cells [2]. Interestingly, it has been suggested that CSCs can explain the great intratumoral heterogeneity of diverse cancers [3]. CSCs exhibit diverse properties that contribute to the progenitor (or stemness) phenotype, tumor progression, and recurrence, including (i) the ability to self-renew, (ii) pluripotency to differentiate into diverse lineages, (iii) tumor initiation, (iv) clonal long-term repopulation potential, and (v) uncontrolled tumor growth capacities that promote metastasis and recurrence [4,5]. Moreover, CSCs are more resistant to chemotherapeutic and radiation therapies; thus, they are considered attractive therapeutic targets in the development of novel pharmacological agents for cancer [1,6]. It has been postulated that during tumorigenesis, epigenetic and genetic alterations may deregulate the expression of multiple protein-encoding genes and non-coding RNAs impacting cancer signaling pathways, resulting in a cell phenotype similar to that of stem cells [7]. Interestingly, an outstanding study reported that glioma CSCs display several properties relevant to endothelial vessel formation, indicating a functional link between cancer cell stemness and angiogenesis [8]. In fact, they have the potential for differentiation into endothelial lineages, suggesting that, to some extent, the vascular endothelium has a tumoral origin, resulting in greatly vascularized tumors that show regions of vasculogenic mimicry [8].

Tumoral angiogenesis is the classical mechanism involved in new vessel formation, which has the function to nourish cancer cells and tissues driving tumor growth and progression [9]. Therefore, the study of interactions and crosstalk between cancer cells and the vasculature and related structures may improve our knowledge, help identify potential gene targets, and develop novel anti-angiogenic strategies in cancer therapy [9,10]. Remarkably, several studies have reported a functional link between angiogenesis and CSCs through the overexpression of vascular endothelial growth factor (VEGF), which is induced by tumoral hypoxia [11]. Recently, a novel mechanism for blood supply has been described in aggressive tumors and denoted as vasculogenic mimicry (VM). This is a microvascularization system in which tumor cells resemble the behavior of endothelial cells, forming de novo channel-like structures themselves [12,13]. Both mechanisms operate in a coordinated way to mediate increased resistance to anti-angiogenic therapies [12]. VM channels, in a surrogate way, promote nutrient acquisition correlating with metastasis and poor prognosis [13]. The VM mechanism may occur in a coordinated way with classical angiogenesis, creating mosaic vessels (a mixture of VM channels and blood vessels), or it may appear alone as a secondary pathway for blood supply when angiogenesis is inhibited by anti-angiogenic drugs [14]. Therefore, further insights into the cellular signaling that triggers VM and angiogenesis in CSCs could help to boost cancer therapies. Recent studies have shown that CSCs and epithelium-to-endothelium transition (EET) accelerate VM formation by stimulating cancer cell plasticity and attaching VM tubules to endothelial vessels [15]. Together, these findings suggest a biological link between CSCs, angiogenesis, and VM; however, both its importance in breast cancer and its regulation by microRNAs are poorly understood.

MicroRNA-204-5p (miR-204) is a tumor suppressor found to be frequently downregulated in diverse types of human cancers, including breast cancer, and it has been associated with tumor progression and poor prognosis [16]. miR-204 exhibits potent anti-tumor effects including the inhibition of cell proliferation, migration, metastasis, and resistance to therapy in multiple tumors [16]. In the present investigation, we analyzed the role of miR-204 in the CSCs (CD44+/CD24−) isolated from MDA-MB-231 and Hs-578t breast cancer cell lines on VM and angiogenesis. Our data showed a prominent role of miR-204 in both cellular processes, suggesting that it can represent a therapeutic target in CSCs and potentially improve the effectiveness of therapies for triple-negative breast cancers.

## 2. Results

### 2.1. Isolation of CD44+/CD24− Subpopulations from MDA-MB-231 and Hs-578t Triple-Negative Breast Cancer Cells

To initiate the study of miR-204 on stemness properties, we first isolated the cancer stem-like cell (CSC) subpopulations from two triple-negative breast cancer cells using flow cytometry and specific antibodies. The expression profile of CD44+/CD24− proteins is widely accepted as a molecular marker of breast cancer stem-like cells [3,17,18]. Indeed, a subpopulation with CD44+ and CD24−/lower phenotype showed increased metastasis to bone in a mouse model of human breast cancer stem-like cells [19]. Therefore, here, we used anti-CD44-PE and anti-CD24-FITC monoclonal antibodies to isolate the CSCs, as described in the Materials and Methods section. Our data indicated that 86.1% of the mesenchymal type MDA-MB-231 cancer cells showed the CD44+/CD24− immunophenotype representative of the subpopulations of CSCs (Figure 1A–C). The percentages of the surface markers in the associated subpopulations were as follows: CD24−/CD44− (12.1%), CD24+/CD44+ (1.8%), and CD44−/CD24+ (0.003%). Likewise, 92% of the Hs-578t cancer cells showed the CD44+/CD24− immunophenotype, whereas the CD24−/CD44−, CD24+/CD44+, and CD44−/CD24+ subpopulations constituted 7.7%, 0.2%, and 0%, respectively (Figure 1D–F). The CD44+/CD24− cells were selected for downstream analyses.

### 2.2. MicroRNA-204 Impairs Mammosphere Formation in Cancer Stem-like Cells

MicroRNAs are negative regulators of the expression of genes participating in the establishment and maintenance of the stemness phenotype; however, a small number of studies outline the functional role of these tiny non-coding RNAs in the pathogenesis of breast cancer CSCs [20,21]. To evaluate the effects of the tumor suppressor miR-204 on the stemness properties of CSCs, we transfected the miR-204 precursor (Figure 2A) and determined the effects on mammosphere formation. No significant differences in the cell viability of miR-204 transfected and control CSCs cells were found (Figure 2B). As expected, non-transfected control and negative control scramble-transfected CD44+/CD24− cells showed the ability to form mammospheres in vitro (Figure 2C). In contrast, the data showed that miR-204 transfection resulted in a significant decrease in the size and number of spheroids during the first and second generations of cell cultures, suggesting an inhibitory effect on the renewal capacities of the CSCs (Figure 2D–F). Moreover, cell-sorting assays on the CSCs showed that miR-204 expression was able to reduce the percentage of CD44+ cells from the CD44+/CD24− subpopulation (Figure 2G,H). These data suggested that miR-204 was able to inhibit some of the stemness properties of the CSCs.

### 2.3. MicroRNA-204 Impairs the Angiogenic Potential of MDA-MB-231 and Hs578t Breast CSCs

The stemness properties of CSCs are involved in the activation of angiogenesis, drug resistance, and metastasis, promoting tumor progression [22,23]. Next, we wondered whether miR-204 has a functional role in angiogenesis in the subpopulations of breast CSCs. The miR-204 precursor was transfected for 48 h, and as controls, non-transfected cells and cells transfected with scramble negative control were used (Figure 3A–D). Then, the vessel generation assays were carried out with CSCs in co-culture with human umbilical vein endothelial cells (HUVEC) for 24 h and the establishment of vessel-like structures was imaged. The data showed that the HUVEC cells in co-culture with non-transfected, mock, and scramble-treated control cancer cells efficiently generated tubular structures (Figure 3A–C). In contrast, the formation of vessel networks was abrogated by miR-204, which resulted in the decrease in the number of tubular structures affecting the general process of HUVEC cells elongation and reorganization (Figure 3D). To better corroborate the anti-angiogenic effect of miR-204, the number of polygons, sprouting cells, connected cells, and number of tubules and nodes were quantified as described [24]. The results showed a significant diminution in the numbers of polygons, sprouting cells, tubules, and nodes affecting the angiogenic index in comparison with the controls (Figure 3E–J).

Likewise, the formation of tubular networks was impaired by miR-204 in Hs-5878t breast CSCs (Figure 4A–D). The data indicated that transfection of miR-204 precursor resulted in a significant diminution in the numbers of polygons, sprouting cells, connected cells, tubules, and nodes affecting the angiogenic index relative to controls (Figure 4E–J).

### 2.4. MicroRNA-204 Inhibits Vasculogenic Mimicry in Breast CSCs

Previous studies suggested that breast cancer stem-like cells could promote VM and metastasis [25,26]; however, the molecular mechanisms and the role of microRNAs regulating the crosstalk between these cellular processes are poorly understood. Therefore, we wondered whether miR-204 has a functional role in VM formation in MDA-MB-231 and Hs-578t cancer stem-like cells (CD44+/CD24−). We used a three-dimensional (3D) cell culture model using a commercial Matrigel enriched in matrix extracellular proteins as a scaffold. First, the precursor miR-204 was transfected for 48 h. As controls, we used non-transfected and scramble-treated cells (Figure 5A). Then, miR-204-expressing cells and controls were grown in hypoxic conditions for 48 h, and the formation of channel-like structures indicative of the initial stages of VM were imaged at 0, 3, 6, and 24 h. The results showed that hypoxia-induced 3D channel-like development was efficiently generated in the control cells. An increase in the number of branch points and capillary-like tubes over time (0–24 h) was found in both MDA-MB-231 and Hs-578t cancer stem-like cells (Figure 5A–C and Figure 6A–C). In contrast, the restoration of miR-204 resulted in a dramatic inhibition of VM upon hypoxia in both CSCs cell lines (Figure 5A–C and Figure 6A–C). A dramatic reduction in the quantity of the branch points and channels compared to the control cells was observed from the earliest time point until 24 h of the channel-like formation assay (Figure 5A–C and Figure 6A–C).

### 2.5. MicroRNA-204 Downregulates the β-Catenin and VEGFA Proteins in MDA-MB-231 Cancer Stem-like Cells

Deregulated Wnt/β-catenin signaling has been associated with the development and survival of CSCs in diverse types of solid tumors [27,28]. In addition, it is well known that vascular endothelial growth factor A (VEGFA) protein function is critical for angiogenesis and VM formation, influencing the pathogenesis of breast cancer [29,30]. Therefore, we decided to evaluate the expression levels of β-catenin and VEGFA proteins in the breast cancer stem-like cells transfected or not with miR-204 mimics. Western blot assays using specific antibodies showed that the expression of the β-catenin protein was significantly decreased in the CSCs transfected with precursor miR-204 in comparison with the non-transfected, mock, and scramble-transfected control cells (Figure 7A,B). Likewise, the expression of the VEGFA protein was significantly downregulated in the CSCs transfected with miR-204 in comparison with the control cells (Figure 7A,C). The levels of GADPH protein analyzed as an endogenous control did not show significant differences between the treated and control cells.

## 3. Discussion

Tumors activate multiple signaling pathways to regulate angiogenesis and VM for efficient tumor progression. Breast cancer stem-like cells show stemness properties that are involved in malignant progression through the activation of tumor renewal, drug resistance, angiogenesis, and metastasis [22,23]. In addition, recent studies have demonstrated that microRNAs have a regulatory role in the expression of genes involved in breast cancer development and progression. During breast tumorigenesis, many microRNAs are aberrantly regulated to promote tumor cell survival. Previously, we, and others, reported that miR-204 is a tumor suppressor with prominent roles in the pathogenesis of breast cancer [31,32,33]. Here, we have extended our previous findings to breast cancer stem-like cells and studied the functions of miR-204 on angiogenesis and VM. Whether miR-204 contributes to the development of angiogenesis and VM in CSCs is poorly understood. It has been reported that CD44+ and CD24 are the major CSC markers described for invasive breast cancer cells from cell lines [19]. Indeed, a subpopulation of CD44+/CD24− phenotypes showed increased aggressiveness and metastasis to human bone in a mouse model of breast CSCs [19]. In addition, a study indicated that CD44(+)/CD24(−/low) breast cancer stem-like cells play a pivotal role associated with the clinical behavior of triple-negative breast cancer [34]. These findings point to the identification of novel therapeutic targets directed to CSCs to improve the effectiveness of current therapies for triple-negative breast cancers. Inspired by these reports, in this study, we used CD44+/CD24− markers to isolate CSCs and studied the role of miR-204 in several cancer hallmarks related to the stemness phenotype. Our data indicated that MDA-MB-231 and Hs-578t cells showed high percentages of CD44+/CD24− cells, in agreement with previous studies reporting 85–99% of CSCs in breast cancer cell lines [34,35,36]. In addition, miR-204 was able to inhibit the stemness properties of CSCs, angiogenesis, and VM that are related to aggressiveness and increased metastasis [36]. Recent evidence indicates that CSCs are involved in VM formation and angiogenesis through epithelial–mesenchymal transition (for a review, see [37]). These findings are relevant, as they allow us to propose that a combination of targeting both CSCs and non-CSCs within the bulk tumor using miR-204 mimics may be beneficial for anti-angiogenic and VM-directed therapies. However, the utilization of mouse models to corroborate this hypothesis is needed for future studies.

It has been reported that Wnt/β-catenin signaling is associated with the survival of CSCs [27,28]. Moreover, VEGFA expression is critical for angiogenesis and vasculogenic mimicry formation in breast cancer [29,30]. We propose that miR-204 is able to downregulate the levels of both β-catenin and VGFA proteins through an indirect mechanism as their corresponding mRNAs have no predicted miR-204 binding sites. Previously, we have reported that TGBR2, PI3K, SRC, FAK, and HIF1-α were direct targets of miR-204 in breast cancer cells [33]. Also, we have reported diverse non-direct targets downregulated by miR-204 including AKT, MEK, and p38 MAPK, signaling transducers which explains, at least in part, its functions in cell migration, proliferation, VM, and angiogenesis in breast cancer cells [33]. Regarding the regulation of β-catenin and VEGFA by miR-204, previously it was reported that miR-204 could regulates HIF1 in breast cancer cells [33], and lung cancer cells [38]. Also, a regulatory loop between miR-204/Sam68/β-catenin was recently reported in breast cancer stem-like cells [39]. In the present study, we propose that miR-204 could downregulates its direct target HIF1-α in response to hypoxia which in turns deregulates VEGFA (a HIF1-α target gene) impacting angiogenesis and VM in breast CSCs. Likewise, as reported by Wang et al. (2015), miR-204 may regulate β-catenin by an indirect mechanism, thus impacting the cell renewal of HER2+ and luminal breast CSCs [39]. Here, we provide evidence at protein-level suggesting that both β-catenin and VEGFA were also downregulated by miR-204 in breast CSCs. The downregulation of β-catenin and VEGFA expression may, at least in part, explain the inhibition of angiogenesis and VM in the breast CSCs expressing miR-204.

The limitations of our study include (i) the lack of evidence regarding the identification of novel miR-204 target genes regulating the expression of β-catenin and VEGFA proteins, and (ii) the need for in vivo studies to evaluate the potential of miR-204 in preclinical models. At the same time, these limitations constitute a guide for future studies focused on the analysis of the molecular mechanisms of miR-204 in vascularization and related processes in CSCs. In conclusion, altogether, our data suggest that breast cancer stem-like cells are targeted by miR-204, which could be potentially translated for therapy. This has implications in the discovery of molecular targets in breast CSCs that may lead to a reduction in the treatment of triple-negative breast cancers.

## 4. Materials and Methods

### 4.1. Cell Cultures

MDA-MB-231 and Hs-578t breast cancer cells were purchased from the American Type Culture Collection (ATCC), and routinely grown in Dulbecco’s modified Eagle’s minimal medium (DMEM-F12) supplemented with 10% fetal bovine serum and the antibiotics penicillin and streptomycin (50 unit/mL; Invitrogen, Carlsbad, CA, USA) at 37 °C in an incubator with a 5% CO_2_ atmosphere.

### 4.2. Isolation of CD44+/CD24− Breast Cancer Stem-like Cells (CSCs)

First, 1 × 10^6^ MDA-MB-231 and Hs-578t CSCs cells were washed with 500 µL of PBS/EDTA/BSA solution twice, and then harvested with 0.05% trypsin/0.025% EDTA. The detached cells were suspended in 100 µL PBS/EDTA/BSA solution. Then, 2 µL of human FCR blocking reagent was added and the cells incubated for 10 min at 4 °C. Then, a combination of fluorochrome-conjugated monoclonal antibodies obtained from Miltenyi Biotec against CD44 (Clone BJ18) and CD24 (Clone 32D12) or their respective IgG1 isotype controls were added to the cell suspension and incubated at 4 °C in the dark for 10 min. Then, the labeled cells were washed in PBS/EDTA/BSA solution, fixed in PBS containing 1% paraformaldehyde, and finally examined using FACS equipment (BD Influx BD Biosciences, Franklin Lakes, NJ, USA).

### 4.3. Mammosphere Formation Assays

A total of 250,000 MDA-MB-231 cells/mL were seeded into a low-adherence 6-well flask (Corning, Glendale, AZ, USA) in serum-free DMEM/F12 media supplemented with 10 ng/mL basic fibroblast growth factor (b-FGF), 20 ng/mL epidermal growth factor (EGF), and 2% B27. These proteins were added to media each 3 days. Mammospheres were formed as non-adherent cells, and the first generation of cells was collected by centrifugation and dissociated to single cells by treatment with 0.05% EDTA/trypsin. Mammospheres were passaged for 2 weeks and imaged on days 0, 6, and 12.

### 4.4. MicroRNA-204 Mimic Transfection

A microRNA-204 precursor and scramble negative control were both transfected at 30 nM or 60 nM in MDA-MB-231 or Hs-578t breast CSCs using lipofectamine 2000. To carry out the transfections, two independent mixtures were made for each well. The first one contained 127.5 µL Optimem (Gibco, Waltham, MA, USA) and 7.65 µL of miR-204 mimics or scramble control (AM17110; Thermo Fisher Scientific, Inc. Walthman, MA, USA). For the mixture, 7.65 µL of lipofectamine 2000 (Thermo Fisher Scientific, Inc.) and 127.5 µL of Optimem were added. The samples were incubated for 10 min at room temperature. Subsequently, they were incubated for 20 min at room temperature. The expression of ectopic miR-204 was confirmed by stem-loop TaqMan quantitative RT-PCR at 48 h post-transfection using total RNA.

### 4.5. Angiogenesis Assays

In vitro angiogenesis assays were carried out on 96-well plates (Costar, Corning Inc., Glendale, AZ, USA) Firstly, the plates were covered with 50 μL BD, Waltman, MA, USA Matrigel (8 mg/mL) and then incubated at 37 °C for 1 h. MDA-MB-231 or Hs-578t breast CSCs (30,000 cells/well) were transfected with miR-204 (30 nM) for 48 h, and subsequently were mixed with HUVECs (30,000 cells/well) and co-cultured on Matrigel. The mock conditions include only culture media, empty lipofectamine, and water. The plates were incubated at 37 °C in a humid atmosphere containing 5% CO_2_ and 95% air for 24 h. The cells were fixed and subsequently stained using 0.5% crystal violet in a 50% ethanol/PBS solution containing 1.25% paraformaldehyde. Images were recorded using an inverted optical microscope. The angiogenic index was calculated using the following formula [24]:Angiogenic score = [(No. sprouting cells)1 + (No. connected cells)2 + (Number of polygons)3] + [0, 1 or 2]/Total number of cells(1)

### 4.6. Vasculogenic Mimicry Assays

Vasculogenic mimicry assays were carried out on three-dimensional (3D) cell cultures using commercial Matrigel enriched in laminin, collagen IV, entactin, and heparin sulfate proteoglycans as a scaffold. MDA-MB-231 or Hs-578t breast CSCs (1 × 10^4^ cells/well) were treated with pre-miR-204 (30 nM) or scramble (30 nM), as previously described [31,32]. The mock conditions include only culture media, empty lipofectamine, and water. Then, the cells were cultured in a 96-well plate covered with Matrigel (50 µL). Subsequently, the cells were incubated at 37 °C in a 5% CO_2_ atmosphere in normoxic (20% O_2_) or hypoxic conditions (1% O_2_) for 48 h. The formation of 3D channel-like structures was carried out by culturing the cells on Matrigel for 0, 3, 6, 9, and 12 h. Images were taken at each time point using an inverted microscope. The data were expressed as mean ± SD. *p* < 0.05 was considered statistically significant.

### 4.7. Western Blots

Total protein extracts were isolated from MDA-MB-231 breast CSCs using a lysis buffer TNTE (50 mM Tris–HCl (pH 7.4), 150 mM NaCl, 0.1% Triton X-100, 1 mM EDTA) supplemented with 2 µL protease inhibitor cocktail and phosphatase 100X (Halt Protease and Phosphatase Inhibitor Cocktail 100×, Thermo Scientific, Waltham, MA, USA). The proteins (60 µg) were separated on 10% SDS-PAGE and transferred to nitrocellulose membrane (Bio-Rad, Hercules, CA, USA). Afterward, the membranes were blocked for 60 min at room temperature with TBST-1X (137 mM NaCl, 20 mM Tris, 0.1% Tween-20, at pH 7.6) containing 5% BSA (Sigma-Aldrich, St. Louis, MI, USA), as previously described [30]. Then, the samples were incubated overnight at 4 °C with mouse anti-GAPDH (1:1000, Santa Cruz Sc47724, Santa Cruz, CA, USA), mouse anti-VEGFA (1:100, Santa Cruz Sc7269), or mouse anti β-catenin (1:200, Santa Cruz Sc7963) primary antibodies. Finally, the membranes were washed 3 times in TBST-1X and incubated with horseradish peroxidase-conjugated goat anti-mouse (1:6000, Jackson Immuno Research JKSN 115-035-003, Cambridge, UK). An ECL detection kit was used to develop the signals. Densitometric analyses of immunodetected bands were performed using the myImage analysis software package.

### 4.8. Statistical Analysis

The data were expressed as mean ± SD of three independent assays. The results were analyzed using one-way ANOVA and Newman–Keuls’s multiple comparison test. A *p* < 0.01 value was considered statistically significant. Analysis was carried out using free GraphPad software version 9.0.

## Figures and Tables

**Figure 1 ncrna-10-00014-f001:**
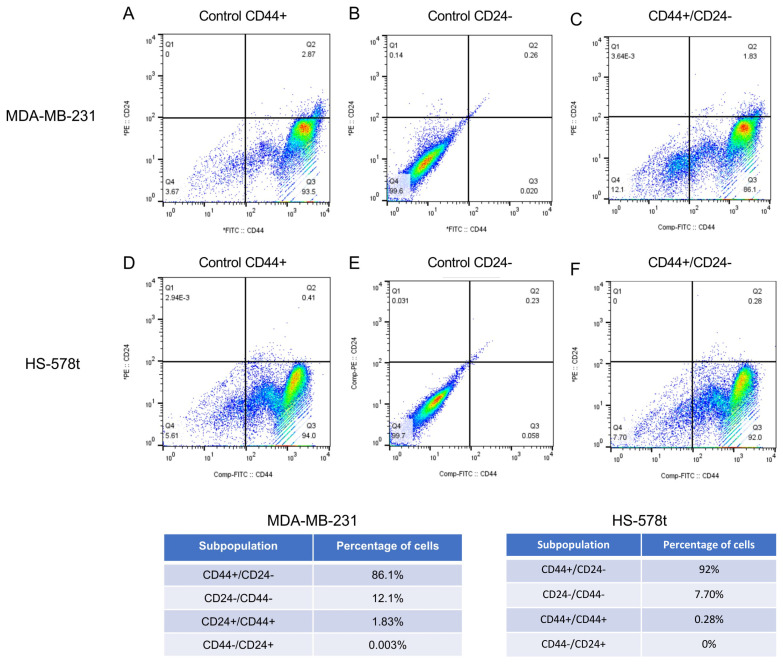
Isolation of MDA-MB-231 and Hs578t breast cancer stem-like cells (CD44+/CD24−). Representative images of fluorescence-activated cell sorting (FACS) analysis using CD44-PE and CD24-FITC antibodies. (**A**) Control MDA-MB-231 cells with CD44, and (**B**) CD24 staining. (**C**) Double staining of CD44 and CD24. (**D**) Control Hs-578t cells with CD44, and (**E**) CD24 staining. (**F**) Double staining of CD44 and CD24. The threshold lines were set according to the isotype control. The reliability of double immunostaining was established by comparison with single immunostaining for CD44 and CD24. Tables show the percentage of cells with specific CD44, CD24, and CD44/CD24 staining in both breast cancer cell lines. Assays were performed in triplicate.

**Figure 2 ncrna-10-00014-f002:**
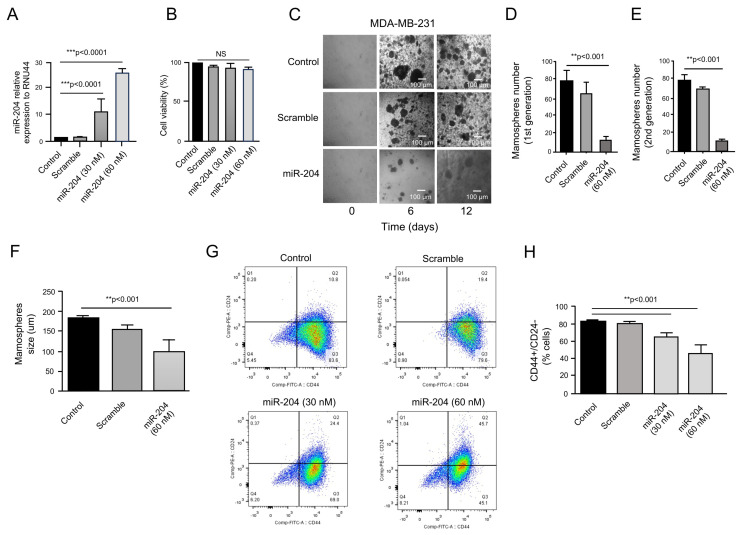
MicroRNA-204 impairs mammosphere formation in CSCs. (**A**) TaqMan quantitative RT-PCR experiments for miR-204 expression in MDA-MB-231 cancer stem-like cells transfected with scramble negative control or precursor miR-204 (30 nM and 60 nM). Data were normalized using RNU44 expression. (**B**) Cell viability assays for MDA-MB-231 cancer stem-like cells transfected with scramble negative control or precursor miR-204 (30 nM and 60 nM). (**C**) Optical microscope images showing CSCs growing as mammospheres during 0, 6, and 12 days of cultivation. Scale bar, 100 μm. (**D**,**E**) Mammospheres counts at 1st and 2nd generation of CSC cultures transfected with scramble negative control or miR-204 mimics. (**F**) Size counts of mammospheres formed by CSCs transfected with scramble or miR-204 on day 6. (**G**) Representative images of FACS analysis using CD44-PE and CD24-FITC antibodies in CSCs. (**H**) Percentage quantification of data in F. NS, non-significant.

**Figure 3 ncrna-10-00014-f003:**
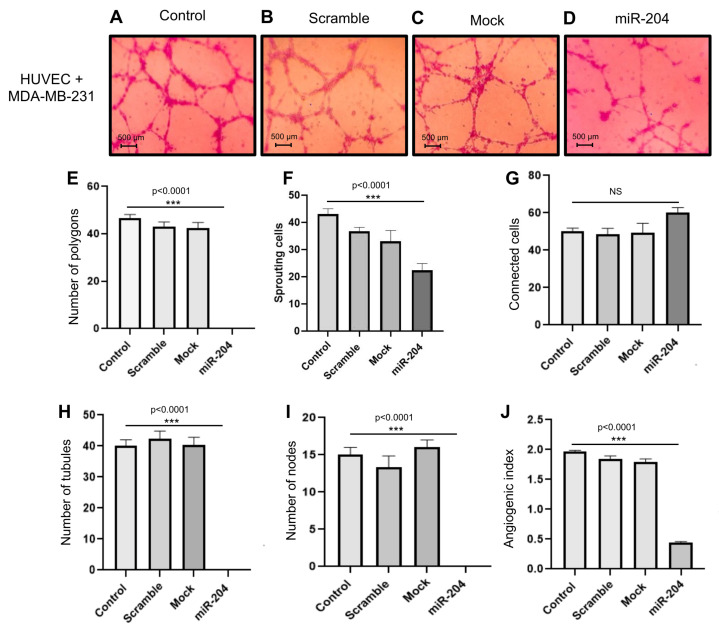
miR-204 mitigates the angiogenesis of HUVEC cells in co-culture with MDA-MB-231 breast cancer stem-like cells. CSCs were plated on Matrigel with DMEM-F12 medium for 24 h and assayed for tubule formation. (**A**) Control, (**B**) scramble, (**C**) mock, and (**D**) miR-204-transfected cells. (**E**–**J**) Graphic representation of the quantification of (**E**) number of polygons, (**F**) number of sprouting cells, (**G**) number of connected cells, (**H**) number of tubules, (**I**) number of nodes, and (**J**) angiogenic index. Assays were performed in triplicate and analyzed with one-way ANOVA.

**Figure 4 ncrna-10-00014-f004:**
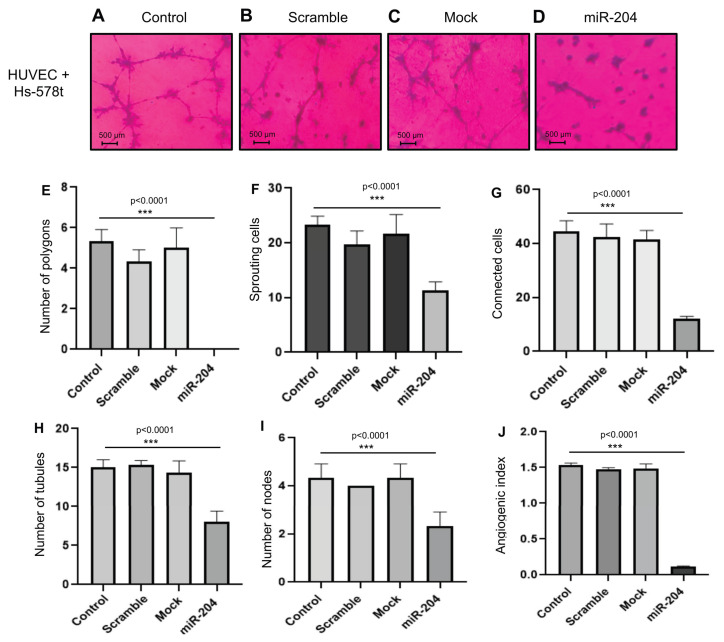
miR-204 inhibits the angiogenesis of HUVEC cells in co-culture with Hs-578t breast cancer stem-like cells. CSCs were plated on Matrigel with DMEM-F12 medium for 24 h and assayed for tubule formation in vitro. (**A**) Control, (**B**) scramble, (**C**) mock, and (**D**) miR-204 cells. (**E**–**J**) Graphic representation of the quantification of (**E**) number of polygons, (**F**) number of sprouting cells, (**G**) number of connected cells, (**H**) number of tubules, (**I**) number of nodes, and (**J**) angiogenic index. Experiments were executed in triplicate and analyzed with one-way ANOVA.

**Figure 5 ncrna-10-00014-f005:**
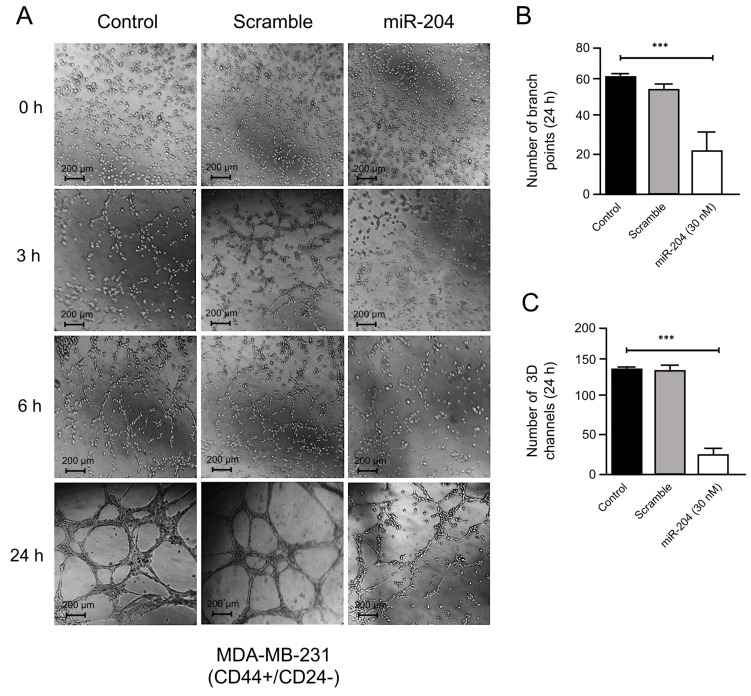
miR-204 impairs vasculogenic mimicry in MDA-MB-231 breast CSCs. Cells were transfected with scramble or miR-204 mimics for 48 h, and then submitted to hypoxia for 48 h and plated on Matrigel for 0–24 h for tubule formation assay. (**A**) Control cells (left panel), scramble (middle panel), miR-204-transfected cells (right panel). (**B**) Number of branch points. (**C**) Number of 3D channels after 24 h. Experiments were performed in triplicate, and data were expressed as mean ± SD. One-way ANOVA. *** *p* < 0.0001.

**Figure 6 ncrna-10-00014-f006:**
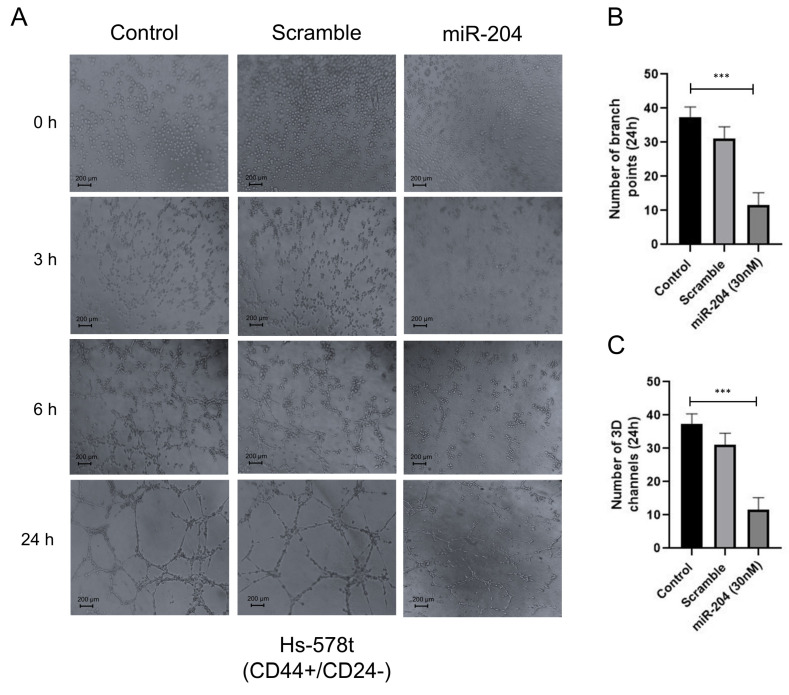
miR-204 inhibits vasculogenic mimicry in Hs-578t breast CSCs. Cells were transfected with scramble or miR-204 mimics for 48 h, submitted to hypoxia for 48 h, and then plated on Matrigel for 0–24 h for tubule formation assay. (**A**) Control cells (left panel), scramble (middle panel), miR-204-transfected cells. (right panel). (**B**) Number of branch points. (**C**) Number of 3D channels after 24 h. Experiments were performed in triplicate, and data were expressed as mean ± SD. One-way ANOVA. *** *p* < 0.0001.

**Figure 7 ncrna-10-00014-f007:**
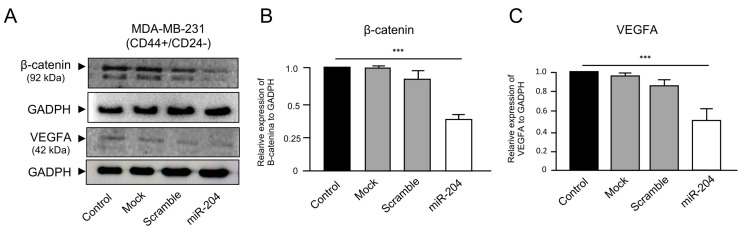
Expression analysis of β-catenin and VEGFA proteins. (**A**) Western blot assays for β-catenin and VEGFA in total protein extracts obtained from control, mock, scramble-transfected, and precursor miR-204-transfected MDA-MB-231 breast cancer stem-like cells. (**B**,**C**) Graphical representation of the quantification of immunodetected bands in A. Experiments were performed in triplicate, and data were expressed as mean ± SD. *** *p* < 0.0001.

## Data Availability

Data are contained within the article.

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
