# Peer review of "MicroRNA-204 Regulates Angiogenesis and Vasculogenic Mimicry in CD44+/CD24− Breast Cancer Stem-like Cells"

_ncrna, 2024, doi:10.3390/ncrna10010014_

Round 1

Reviewer 1 Report

Comments and Suggestions for Authors

In the manuscript by Resendiz-Hernandez et al., focusing on miR-204's role in angiogenesis and vasculogenic mimicry within CD44+/CD24- breast cancer stem-like cells, several areas warrant further refinement for enhanced scientific rigor:

  1. The abstract, particularly lines 20-25, necessitates meticulous language editing to enhance clarity and precision.
  2. What is the basal expression of miR-204 in MDA-MB-231 cells? Quantification using digital PCR will be helpful. This analysis should be extended to include more breast cancer lines.
  3. Besides MDA-MB-231, it is unclear if other breast cancer cell lines were assessed for stem cell-like characteristics. Expanding this analysis could validate the generalizability of the findings across various breast cancer subtypes.
  4. Scale bar for Y axis of Figure 2A is missing.
  5. To substantiate the findings observed in Figure 2 post-miR-204 overexpression in MDA-MB-231 cells, replicating these experiments in a cell line characterized by high endogenous miR-204 expression is recommended. 
  6. In lines 152-166, there is an erroneous reference to Figure 3 as Figure 2. 
  7. Are β-catenin and VEGFA direct targets of miR-204? If yes this needs to be demonstrated using a luciferase assay or by citing existing literature. If not, direct targets of miR-204 should be evaluated to understand the molecular mechanism of regulation of angiogenesis in Breast cancer stem-like cells via miR-204.
Comments on the Quality of English Language

The manuscript needs meticulous english language editing to enhance clarity and precision.

Author Response

Reviewer 1

We acknowledge to editor and reviewer for the opportunity to revise the manuscript. Your critical suggestions that we have fully replied greatly increase the quality of our study. All amendments have been marked in yellow color in the revised text (manuscript with highlighted changes) for your easy reference and reading. We have carefully reviewed the manuscript according to the referee suggestions and provide a point-by-point response.

In the manuscript by Resendiz-Hernandez et al., focusing on miR-204's role in angiogenesis and vasculogenic mimicry within CD44+/CD24- breast cancer stem-like cells, several areas warrant further refinement for enhanced scientific rigor:

The abstract, particularly lines 20-25, necessitates meticulous language editing to enhance clarity and precision.

Reply: Thanks to reviewer for the comments. We have reorganized the abstract with new data and rephrased the lines 20-25. Moreover, the complete text was corrected by a native speaker at MDPI language services.

What is the basal expression of miR-204 in MDA-MB-231 cells? Quantification using digital PCR will be helpful. This analysis should be extended to include more breast cancer lines.

Reply: We really appreciate your wise comments. MiR-204 is a well-known tumor suppressor, and its expression has been previously reported in diverse papers as downregulated in MDA-MB-231, Hs578t and MCF-7 breast cancer cell lines in comparison with normal breast cancer cell lines. Also, we have reported that the expression of miR-204 in tumors is low relative to normal breast tissues (Flores-Perez et al., 2016). Thus, we consider that this point has been well documented in the literature. To clarify this point, we have added a new paragraph with functions of miR-204 in breast cancer (Page 2, lanes 92-96). For your reference, several relevant papers are enlisted below:

  1. Wang X, et al. MicroRNA-204 targets JAK2 in breast cancer and induces cell apoptosis through the STAT3/BCl-2/survivin pathway. International Journal of Clinical and Experimental Pathology. 2015;8(5):5017-5025.
  2. Yang, F., et al (2023). MicroRNA-204-5p: A pivotal tumor suppressor. Cancer medicine, 12(3), 3185–3200.https://doi.org/10.1002/cam4.5077.
  3. Li, W., et al. (2014). Decreased expression of miR-204 is associated with poor prognosis in patients with breast cancer. International journal of clinical and experimental pathology, 7(6), 3287–3292.
  4. Flores-Pérez, A., et al. (2016). Dual targeting of ANGPT1 and TGFBR2 genes by miR-204 controls angiogenesis in breast cancer. Scientific reports, 6, 34504. https://doi.org/10.1038/srep34504.
  5. Hong, B. S., et al (2019). Tumor Suppressor miRNA-204-5p Regulates Growth, Metastasis, and Immune Microenvironment Remodeling in Breast Cancer. Cancer research, 79(7), 1520–1534. https://doi.org/10.1158/0008-5472.CAN-18-0891.

Besides MDA-MB-231, it is unclear if other breast cancer cell lines were assessed for stem cell-like characteristics. Expanding this analysis could validate the generalizability of the findings across various breast cancer subtypes.

Scale bar for Y axis of Figure 2A is missing.

Reply: Thanks to referee for the valuable comments. As the reviewer suggested, we have performed new experiments including an additional breast cancer cell line (Hs-578t) and isolated the corresponding CD44+ subpopulations of CSCs. We have selected this cell line as this subtype showed better abilities to form channels-like structures in vitro (Salinas Vera et al., 2018. Cancer letters, https://doi.org/10.1016/j.canlet.2018.06.003) and to maintain the study focused on triple negative breast cancer. Data indicate that 92% of Hs-578t cells showed the CD44+/CD24- immunophenotype, whereas the sub-populations CD24-/CD44-; CD24+/CD44+, and CD44-/CD24+ constitute the 7.7%, 0.2 % and 0%, respectively (corrected Figure 1D-F). In addition, the effects of miR-204 restoration in Hs-578t cells on angiogenesis and vasculogenic mimicry were further evaluated. Data showed that miR-204 restoration in Hs-578t CSCs was able to inhibit both angiogenesis and vasculogenic mimicry (Pages 6 and 8, respectively).

Finally, as the reviewer suggested the scale bar in Figure 2A has been added.

To substantiate the findings observed in Figure 2 post-miR-204 overexpression in MDA-MB-231 cells, replicating these experiments in a cell line characterized by high endogenous miR-204 expression is recommended.

Reply: It’s an interesting the idea, however no breast cancer cell lines with upregulated expression of miR-204 have been identified. In contrast, all the cell lines studied in diverse papers showed a very low expression of miR-204 relative to “normal” breast cells and tissues.

In lines 152-166, there is an erroneous reference to Figure 3 as Figure 2.

Reply: We have corrected the references to figure 3.

Are β-catenin and VEGFA direct targets of miR-204? If yes this needs to be demonstrated using a luciferase assay or by citing existing literature. If not, direct targets of miR-204 should be evaluated to understand the molecular mechanism of regulation of angiogenesis in Breast cancer stem-like cells via miR-204.

Reply: We really appreciate your feedback. This it’s a very interesting question. Predictions using TargetScan showed that both β-catenin and VEGFA transcripts are not direct targets of miR-204, as they lack bindings sites for miR-204 at their 3´UTRs.

Previously we have reported that TGBR2, PI3K, SRC, FAK and HIF1a are truly miR-204 direct targets in breast cancer cells. Also, we have reported diverse non-direct targets downregulated by miR-204 including AKT, MEK, and p38 MAPK, which explains, at least in part, its functions in cell migration, proliferation, vasculogenic mimicry and angiogenesis in breast cancer cells. Thus, we have advanced in the knowledge of molecular mechanisms regulated by miR-204 in cancer.

Regarding the regulation of β-catenin and VEGFA by miR-204, previously it was reported that miR-204 could regulates HIF1A in breast and lung cancer cells (1,2):

  1. Salinas-Vera, Y.M. 2018. Cooperative multi-targeting of signaling networks by angiomiR-204 inhibits vasculogenic mimicry in breast cancer cells. Cancer letters, 432, 17–27. https://doi.org/10.1016/j.canlet.2018.06.003.
  2. Liu, X.N., et al. 2021. microRNA-204 shuttled by mesenchymal stem cell-derived exosomes inhibits the migration and invasion of non-small-cell lung cancer cells via the KLF7/AKT/HIF-1α axis. Neoplasma, 68(4), 719–731. https://doi.org/10.4149/neo_2021_201208N1328.

Also, a regulatory loop between miR-204/Sam68/β-catenin was reported in breast cancer stem-like cells:

  1. Wang, L. 2015. CONSORT: Sam68 Is Directly Regulated by MiR-204 and Promotes the Self-Renewal Potential of Breast Cancer Cells by Activating the Wnt/Beta-Catenin Signaling Pathway. Medicine, 94(49), e2228. https://doi.org/10.1097/MD.0000000000002228.

In the present study, we propose that miR-204 could downregulates its direct target HIF1A in response to hypoxia which in turns deregulates VEGFA (a HIF1a transcriptional target gene) impacting angiogenesis and vasculogenic mimicry in breast CSCs. Likewise, as reported Wang et al., (2015) in a recent study miR-204 may regulates β-catenin by an indirect mechanism and thus impacting cell renewal of HER2+ and luminal breast CSCs. Here, we provide evidence a protein-level that both β-catenin and VEGFA are regulated by miR-204 in breast CSCs. However, we agree with the reviewer about the lack of novel targets of miR-204 in triple negative breast CSCs which will remains to be discovered in future research.

These comments have been now added to discussion section (Page 10, lanes 284-300). Also, we have remarked the limitation of our study (Page 10, lanes 302-307).

The manuscript needs meticulous english language editing to enhance clarity and precision.

Reply: Thanks to reviewer for the comments. The complete text was corrected by a native speaker at MDPI language services as you suggested.

Reviewer 2 Report

Comments and Suggestions for Authors

The investigators in this manuscript have scrutinized the involvement of miR-204 in angiogenesis and vasculogenic mimicry within cancer stem-like cells in a breast cancer model. While the study provides valuable data contributing to the progression of the field, further investigations are imperative to enhance its methodological robustness. The following recommendations outline the necessary supplementary studies:

1. In the discussion section, the authors posit that miR-204 may regulate CD44 directly through binding (lines 253-254). However, conclusive evidence is lacking, necessitating additional experiments to substantiate the direct or indirect regulatory role of miR-204 on CD44. It is essential to undertake experiments that elucidate the specific interactions between miR-204 and CD44.

2. The discussion suggests that the manuscript's findings support the proposition of targeting both cancer stem cells (CSCs) and non-CSCs using miR-204. Nevertheless, further, rigorous studies are indispensable to substantiate this conclusion. Specifically, experiments elucidating the mechanistic regulation of Wnt/beta-catenin and VEGFA by miR-204 in breast cancer stem-like cells are imperative. The precise regulatory mechanisms governing these proteins by miR-204 need thorough exploration.

3. The manuscript demonstrates that miR-204 modulates vasculogenic mimicry through transient overexpression. To strengthen this evidence, validation in an alternative cell line is warranted. Additionally, conducting reversal experiments will further fortify the reliability of the data, providing a more comprehensive understanding of the regulatory dynamics involved in vasculogenic mimicry by miR-204.

4. Validation of the findings in an in vivo setting is essential for establishing the clinical relevance of the study. Furthermore, an analysis of publicly available datasets will contribute to the broader understanding and generalizability of the observed effects. Integrating these approaches will bolster the credibility and translational potential of the study's outcomes.

5. The discussion needs to be rewritten more comprehensively, right now its looking more like results section.

Minor comments:

11. Y-axis numbers are missing in the Fig.2A.

22. In the Fig.2F bottom right panel has been labeled as miR-204 (30 nM), is that correct?

33. In the discussion line 233-234 doesn’t make any sense.

Author Response

Reviewer 2

We acknowledge to editor and reviewer for the opportunity to revise the manuscript. Your critical suggestions that we have fully replied greatly increase the quality of our study. All amendments have been marked in yellow color in the revised text (manuscript with highlighted changes.doc) for your easy reference and reading. We have carefully reviewed the manuscript according to the referee suggestions and provide a point-by-point response.

The investigators in this manuscript have scrutinized the involvement of miR-204 in angiogenesis and vasculogenic mimicry within cancer stem-like cells in a breast cancer model. While the study provides valuable data contributing to the progression of the field, further investigations are imperative to enhance its methodological robustness. The following recommendations outline the necessary supplementary studies:

  1. In the discussion section, the authors posit that miR-204 may regulate CD44 directly through binding (lines 253-254). However, conclusive evidence is lacking, necessitating additional experiments to substantiate the direct or indirect regulatory role of miR-204 on CD44. It is essential to undertake experiments that elucidate the specific interactions between miR-204 and CD44.

Reply: We appreciate the wise comments. You right about the lack of evidence demonstrating that CD44 is regulated by miR-204. To avoid overinterpretation, we have deleted this conclusion. The validation of CD44 and additional genes involved in stemness phenotype as a novel miR-204 target and its consequences in stemness and multiple cancer hallmarks will be addressed in a future ongoing study.

  1. The discussion suggests that the manuscript's findings support the proposition of targeting both cancer stem cells (CSCs) and non-CSCs using miR-204. Nevertheless, further, rigorous studies are indispensable to substantiate this conclusion. Specifically, experiments elucidating the mechanistic regulation of Wnt/beta-catenin and VEGFA by miR-204 in breast cancer stem-like cells are imperative. The precise regulatory mechanisms governing these proteins by miR-204 need thorough exploration.

Reply: We really appreciate your feedback. This it’s a very interesting question. Predictions using TargetScan showed that both β-catenin and VEGFA transcripts are not direct targets of miR-204, as they lack bindings sites for miR-204 at their 3´UTRs.

Previously, we have reported that TGBR2, PI3K, SRC, FAK and HIF1a are truly miR-204 direct targets in breast cancer cells. Also, we have reported diverse non-direct targets downregulated by miR-204 including AKT, MEK, and p38 MAPK, which explains, at least in part, its functions in cell migration, proliferation, vasculogenic mimicry and angiogenesis in breast cancer cells. Thus, we have advanced in the knowledge of molecular mechanisms regulated by miR-204 in cancer.

Regarding the regulation of β-catenin and VEGFA by miR-204, previously it was reported that miR-204 could regulates HIF1A in breast and lung cancer cells (1,2):

  1. Salinas-Vera, Y.M. 2018. Cooperative multi-targeting of signaling networks by angiomiR-204 inhibits vasculogenic mimicry in breast cancer cells. Cancer letters, 432, 17–27. https://doi.org/10.1016/j.canlet.2018.06.003.
  2. Liu, X.N., et al. 2021. microRNA-204 shuttled by mesenchymal stem cell-derived exosomes inhibits the migration and invasion of non-small-cell lung cancer cells via the KLF7/AKT/HIF-1α axis. Neoplasma, 68(4), 719–731. https://doi.org/10.4149/neo_2021_201208N1328.

Also, a regulatory loop between miR-204/Sam68/β-catenin was reported in breast cancer stem-like cells:

  1. Wang, L. 2015. CONSORT: Sam68 Is Directly Regulated by MiR-204 and Promotes the Self-Renewal Potential of Breast Cancer Cells by Activating the Wnt/Beta-Catenin Signaling Pathway. Medicine, 94(49), e2228. https://doi.org/10.1097/MD.0000000000002228.

In the present study, we propose that miR-204 could downregulates its direct target HIF1A in response to hypoxia which in turns deregulates VEGFA (a HIF1a transcriptional target gene) impacting angiogenesis and vasculogenic mimicry in breast CSCs. Likewise, as reported Wang et al., (2015) in a recent study miR-204 may regulates β-catenin by an indirect mechanism and thus impacting cell renewal of HER2+ and luminal breast CSCs. Here, we provide evidence a protein-level that both β-catenin and VEGFA are regulated by miR-204 in breast CSCs. However, we agree with the reviewer about the lack of novel targets of miR-204 in triple negative breast CSCs which will remains to be discovered in future research.

These comments have been now added to discussion section (Page 10, lanes 284-300). Also, we have remarked the limitations of our study in discussion section (Page 10, lanes 302-307).

  1. The manuscript demonstrates that miR-204 modulates vasculogenic mimicry through transient overexpression. To strengthen this evidence, validation in an alternative cell line is warranted. Additionally, conducting reversal experiments will further fortify the reliability of the data, providing a more comprehensive understanding of the regulatory dynamics involved in vasculogenic mimicry by miR-204.

Reply: Thanks to referee for the valuable comments. As the reviewer suggested, we have performed new experiments including an additional breast cancer cell line (Hs-578t) and isolated the corresponding CD44+ subpopulations of CSCs. We have selected this cell line as this subtype showed better abilities to form channels-like structures in vitro (Salinas Vera et al., 2018. Cancer letters, https://doi.org/10.1016/j.canlet.2018.06.003) and to maintain the study focused on triple negative breast cancer. Data indicate that 92% of Hs-578t cells showed the CD44+/CD24- immunophenotype, whereas the sub-populations CD24-/CD44-; CD24+/CD44+, and CD44-/CD24+ constitute the 7.7%, 0.2 % and 0%, respectively (Figure 1D-F). In addition, the effects of miR-204 restoration in Hs-578t cells on angiogenesis and vasculogenic mimicry were further evaluated. Data showed that miR-204 restoration in Hs-578t CSCs was able to inhibit both angiogenesis and vasculogenic mimicry (Pages 6 and 8, respectively).

Regarding the reversal experiments, it’s an interesting the idea, however no breast cancer cell lines with upregulated expression of miR-204 have been identified. In contrast, all the cell lines reported in diverse papers showed a very low expression of miR-204, thus overexpression is the most reliable approach to study the miR-204 functions in breast cancer.

  1. Validation of the findings in an in vivo setting is essential for establishing the clinical relevance of the study. Furthermore, an analysis of publicly available datasets will contribute to the broader understanding and generalizability of the observed effects. Integrating these approaches will bolster the credibility and translational potential of the study's outcomes.

Reply: We thanks to reviewer for the comments. It’s an interesting idea, however there are technical limitations as no in vivo model for the study of angiogenesis and vasculogenic mimicry on breast CSCs have been described which limits the establishment of the clinical relevance of data. Future research will be focused on the establishment of such in vivo approaches to dilucidated the miR-204 roles on stemness and renewal properties of breast CSCs using mouse models. We have remarked this limitation in our study in discussion section (Page 10, lanes 302-307).

  1. The discussion needs to be rewritten more comprehensively, right now its looking more like results section.

Reply: Discussion section has been rewritten, avoiding repeating the results.

Minor comments:

  1. Y-axis numbers are missing in the Fig.2A.

Reply: The scale bar in corrected Figure 2A has been added.

  1. In the Fig.2F bottom right panel has been labeled as miR-204 (30 nM), is that correct?

Reply: The figure 2F has been corrected as 60 nM.

  1. In the discussion line 233-234 doesn’t make any sense.

Reply: Sentence has been modified and discussion rewritten.

Reviewer 3 Report

Comments and Suggestions for Authors

This manuscript discussed how miR-204 influenced angiogenesis in cancer stem-like cells (CSCs) of breast cancer using in vitro assays. The manuscript is easy to read and understand and most data were clearly presented. To further improve the manuscript and make the presented data more convincing: 

Major concerns:

1. Can the authors briefly discuss miR-204 in their introduction? For example, its biological/cellular function, relevance in cancer, etc.

2. In line 306-308 the authors said the miR-204 expression was confirmed by qPCR. Can the authors include this data in one of the main/supplementary figures?

3. Line 312-313 is confusing. Did the authors mix HUVEC and CSC cells together, and then transfected both cells together with miR-204? Please clarify.

4. For FACS experiments, were live/dead cell stain included in experiments to exclude dead cells from the analysis? I didn't see this process being included in the method section.

5. Please describe how the mammosphere assay was performed in Figure 2. 

6. Have the authors considered the possibility that miR-204 overexpression decreased cell growth and/or caused cell death, which then led to the observed phenotypes (reduced mammosphere growth, reduced angiogenesis and reduced vasculogenic mimicry)? If miR-204 did reduce cell growth, how would that impact the data interpretation and conclusions?

7. All experiments were performed using one breast cancer cell line. Can authors demonstrate the effects of miR-204 on CSC using more CSC breast cancer cell lines? The authors can choose 1-2 assays to repeat.

Minor concerns:

1. Line 70, "Angiogenesis is the classical dependent-endothelial cells mechanism for..." I'm not sure what "dependent-endothelial" means here. Please elaborate or correct.

2. Figure 2E, the mammosphere sizes were quantified at what time point post miR-204 transfection?

3. Figure 2F, the label of the bottom-right FACS plot seems wrong, should it be 60nM instead of 30nM miR-204?

4. In Figure 3 and 5, what is "mock" condition? Please elaborate.

5. Images in Figure 3 and 4 are lacking scale bars.

6. Figure 3E-J have too many colors which can be distractive during scientific presentation. The authors may consider using different colors for different treatment conditions (control-scramble-mock-miR-204), and using this same color scheme throughout all panels. Since the metrics are indicated by the y axis of different plots, it's not necessary to distinguish them using different colors.

7. The scientific notations were not properly labeled (e.g. line 290 1x106 cells; line 332 1x104 cells/well). Please correct.

Comments on the Quality of English Language

The academic writing was proper and easy to read overall, with some minor mistakes. Specifically:

1. The spelling of "mamosphere" seems to be "mammosphere" instead (e.g. line 30). 

2. Line 61, "Mechanistically it’s have been postulated that..." -> it has been postulated.

3. Line 87, "VM mechanism may occurs..." -> may occur.

4. Line 97, "and its regulation by microRNAs its poorly understood." -> is poorly understood. 

5. Line 201, "brach points" -> branch points.

6. Line 290, "and the harvested with ..." -> and then harvested.

Author Response

Reviewer 3

We acknowledge to editor and reviewer for the opportunity to revise the manuscript. Your critical suggestions that we have fully replied greatly increase the quality of our study. All amendments have been marked in yellow color in the revised text (manuscript with highlighted changes.doc) for your easy reference and reading. We have carefully reviewed the manuscript according to the referee suggestions and provide a point-by-point response.

This manuscript discussed how miR-204 influenced angiogenesis in cancer stem-like cells (CSCs) of breast cancer using in vitro assays. The manuscript is easy to read and understand and most data were clearly presented. To further improve the manuscript and make the presented data more convincing:

Major concerns:

  1. Can the authors briefly discuss miR-204 in their introduction? For example, its biological/cellular function, relevance in cancer, etc.

Reply: We really appreciate your feedback. As the reviewer suggested, we have added a paragraph with the general miR-204 functions in cancer and added a relevant paper review as reference (Page 2, lanes 92-96).

  1. In line 306-308 the authors said the miR-204 expression was confirmed by qPCR. Can the authors include this data in one of the main/supplementary figures?

Reply: Thanks for the comments. The qRT-PCR results for miR-204 restoration in cancer cells are showed in the corrected figure 2A.

  1. Line 312-313 is confusing. Did the authors mix HUVEC and CSC cells together, and then transfected both cells together with miR-204? Please clarify.

Reply: We apologize for the mistake. You right about the sentence which is confusing. We have clarified the method and rewritten the sentence as follows: “MDA-MB-231 or Hs-578t breast CSCs (30,000 cells/well) were transfected with miR-204 (30 nM) for 48 h and subsequently were mixed with HUVECs (30,000 cells/well) and co-cultured on Matrigel.” (Page 11, lanes 350-352).

  1. For FACS experiments, were live/dead cell stain included in experiments to exclude dead cells from the analysis? I didn't see this process being included in the method section.

Reply: After cell sorting, the CD44 cells were seeded and twice passaged before further experiments, thus dead cells were excluded for all the analyses.

  1. Please describe how the mammosphere assay was performed in Figure 2.

Reply: We apologize for the omission. As reviewer suggested, we have added a new paragraph describing the methodology as follows: “4.3. Mammospheres formation assays. 250,000 MDA-MB-231 cells/mL were seeded into a low-adherence 6-wells flask (Corning) in serum-free DMEM/F12 media supplemented with 10 ng/mL basic fibroblast growth factor (b-FGF), 20 ng/mL epidermal growth factor (EGF), and 2% B27. These proteins were added to media each 3 days. Mammospheres were formed as non-adherent cells, and the first generation of cells was collected by centrifugation and dissociated to single cells by treatment with 0.05% EDTA/trypsin. Mammospheres were passaged for 2 weeks and imaged on the days 0, 6 and 12.” (Page 10-11, lanes 329-336).

  1. Have the authors considered the possibility that miR-204 overexpression decreased cell growth and/or caused cell death, which then led to the observed phenotypes (reduced mammosphere growth, reduced angiogenesis and reduced vasculogenic mimicry)? If miR-204 did reduce cell growth, how would that impact the data interpretation and conclusions?

Reply: We thanks to reviewer for address this important point. We previously have checked the impact of miR-204 transfection on cell viability, and we did not find significant differences on cell viability of miR-204-transfected cells relative to controls, these data were now added to Figure 2B.

  1. All experiments were performed using one breast cancer cell line. Can authors demonstrate the effects of miR-204 on CSC using more CSC breast cancer cell lines? The authors can choose 1-2 assays to repeat.

Reply: Thanks to referee for the valuable comments. As the reviewer suggested, we have performed new experiments including an additional breast cancer cell line (Hs-578t) and isolated the corresponding CD44+ subpopulations of CSCs. We have selected this cell line as this subtype showed better abilities to form channels-like structures in vitro (Salinas Vera et al., 2018. Cancer letters, https://doi.org/10.1016/j.canlet.2018.06.003) and to maintain the study focused on triple negative breast cancer. https://doi.org/10.1016/j.canlet.2018.06.003). Data showed that 92% of Hs-578t cells showed the CD44+/CD24- immunophenotype, whereas the sub-populations CD24-/CD44-; CD24+/CD44+, and CD44-/CD24+ constitute the 7.7%, 0.2 % and 0%, respectively (Figure 1D-F). In addition, the effects of miR-204 restoration in Hs-578t cells on angiogenesis and vasculogenic mimicry were further evaluated. Data showed that miR-204 restoration in Hs-578t CSCs was able to inhibit both angiogenesis and vasculogenic mimicry (Pages 6 and 8, respectively).

Minor concerns:

  1. Line 70, "Angiogenesis is the classical dependent-endothelial cells mechanism for..." I'm not sure what "dependent-endothelial" means here. Please elaborate or correct.

Reply: We have corrected sentence as follows: “Tumoral angiogenesis is the classical mechanism involved in new vessel formation, which has the function to nourish cancer cells and tissues driving tumor growth and progression [9].” (Page 2, lane 69-71).

  1. Figure 2E, the mammosphere sizes were quantified at what time point post miR-204 transfection?

Reply: Mamospheres count was performed at day 6. This note has been added to figure 2 legend (Page 5, lane 153).

  1. Figure 2F, the label of the bottom-right FACS plot seems wrong, should it be 60nM instead of 30nM miR-204?

Reply: Thanks to reviewer for the observation. You right…the legend is incorrect. We have now corrected the label in the figure 2G.

  1. In Figure 3 and 5, what is "mock" condition? Please elaborate.

Reply: We apologize for the omission. The mock condition includes only culture medium, empty lipofectamine, and water. This information has been now added to methods sections for 4.4 Angiogenesis (Page 11, lanes 352-353) and 4.5 Vasculogenic mimicry assays (Page 11, lanes 363-364).

  1. Images in Figure 3 and 4 are lacking scale bars.

Reply: Scale bars have been added to figures 3-6.

  1. Figure 3E-J have too many colors which can be distractive during scientific presentation. The authors may consider using different colors for different treatment conditions (control-scramble-mock-miR-204), and using this same color scheme throughout all panels. Since the metrics are indicated by the y axis of different plots, it's not necessary to distinguish them using different colors.

Reply: Its true, I think the same the image its visually very heavy. As you suggested, we have edited the images and changed the colored bars by gray scale.

  1. The scientific notations were not properly labeled (e.g. line 290 1x106 cells; line 332 1x104 cells/well). Please correct.

Reply: The sentence has been corrected as you suggested, thanks. (Page 10, lane 319).

The academic writing was proper and easy to read overall, with some minor mistakes. Specifically:

Reply: Thanks to reviewer for the comments. The complete text was now corrected by a native speaker at MDPI language services.

  1. The spelling of "mamosphere" seems to be "mammosphere" instead (e.g. line 30).

Reply Thanks for the observation. We have changed “mamosphere” by “mammospheres”. (Page 1, lane 32; Page 4 lanes 128, 134 and 145).

  1. Line 61, "Mechanistically it’s have been postulated that..." -> it has been postulated.

Reply: We have shortened the sentence as suggested (Page 2, lane 60).

  1. Line 87, "VM mechanism may occurs..." -> may occur.

Reply: We have corrected the word as you suggested (Page 2, lane 82).

  1. Line 97, "and its regulation by microRNAs its poorly understood." -> is poorly understood.

Reply: The word has been corrected (Page 2, lane 91).

  1. Line 201, "brach points" -> branch points.

Reply: The word has been corrected (Page 7, lane 211).

  1. Line 290, "and the harvested with ..." -> and then harvested.

Reply: The word has been corrected (Page 10, lane 320).

Round 2

Reviewer 1 Report

Comments and Suggestions for Authors

The comments have been sufficiently addressed and substantial changes have been made to the manuscript. 

Author Response

Reply: We appreciate the last comments of reviewer. Thanks for the revision and feedback.

Reviewer 2 Report

Comments and Suggestions for Authors

I would like to thanks the authors for addressing most of the  of the suggestions. this has improved the manuscript significantly.

Author Response

(The authors gave the same response as above.)

Reviewer 3 Report

Comments and Suggestions for Authors

The authors have successfully addressed all my concerns in the revised manuscript, except that the scale bar labels in Figure 3-6 should be corrected: "uM" should be changed to "um". Otherwise, the manuscript is clearly presented and suitable for publication.

Author Response

Reply: We appreciate the last comments. As the reviewer suggested we have corrected the legend uM in figures 3-6.